# Assessment of the Impact of Carvedilol Administered Together with Dexrazoxan and Doxorubicin on Liver Structure and Function, Iron Metabolism, and Myocardial Redox System in Rats

**DOI:** 10.3390/ijms25042219

**Published:** 2024-02-13

**Authors:** Jaroslaw Szponar, Agnieszka Gorska, Marta Ostrowska-Lesko, Agnieszka Korga-Plewko, Michal Tchorz, Erwin Ciechanski, Anna Dabrowska, Ewa Poleszak, Franciszek Burdan, Jaroslaw Dudka, Marek Murias, Slawomir Mandziuk

**Affiliations:** 1Toxicology Clinic, Faculty of Medicine, Medical University of Lublin, 100 Krasnik Avenue, 20-550 Lublin, Poland; jaroslaw.szponar@umlub.pl (J.S.); agnieszka.gorska@umlub.pl (A.G.); michal.tchorz@umlub.pl (M.T.); 2Clinical Department of Toxicology and Cardiology, Stefan Wyszynski Regional Specialist Hospital, 100 Krasnik Avenue, 20-550 Lublin, Poland; 3Department of Toxicology, Medical University of Lublin, 8b Jaczewski Street, 20-090 Lublin, Poland; erwin.ciechanski@umlub.pl (E.C.); jaroslaw.dudka@umlub.pl (J.D.); 4Independent Medical Biology Unit, Medical University, 8b Jaczewski Street, 20-090 Lublin, Poland; agnieszka.korga-plewko@umlub.pl; 5Clinical Department of Cardiology, Stefan Wyszynski Regional Specialist Hospital, 100 Krasnik Avenue, 20-550 Lublin, Poland; 6Department of Applied Pharmacy, Medical University of Lublin, 1 Chodźko Street, 20-093 Lublin, Poland; ewa.poleszak@umlub.pl; 7Human Anatomy Department, Medical University of Lublin, 4 Jaczewski Street, 20-090 Lublin, Poland; franciszek.burdan@umlub.pl; 8Department of Toxicology, Poznan University of Medical Sciences, 3 Rokietnicka Street, 60-608 Poznan, Poland; marek.murias@ump.edu.pl; 9Department of Pneumology, Oncology and Allergology, Medical University of Lublin, 8 Jaczewski Street, 20-090 Lublin, Poland; slawomir.mandziuk@umlub.pl

**Keywords:** post-anthracycline cardiotoxicity, cardioprotection, dexrazoxane, carvedilol, doxorubicin, cardiac molecular metabolism

## Abstract

Late cardiotoxicity is a formidable challenge in anthracycline-based anticancer treatments. Previous research hypothesized that co-administration of carvedilol (CVD) and dexrazoxane (DEX) might provide superior protection against doxorubicin (DOX)-induced cardiotoxicity compared to DEX alone. However, the anticipated benefits were not substantiated by the findings. This study focuses on investigating the impact of CVD on myocardial redox system parameters in rats treated with DOX + DEX, examining its influence on overall toxicity and iron metabolism. Additionally, considering the previously observed DOX-induced ascites, a seldom-discussed condition, the study explores the potential involvement of the liver in ascites development. Compounds were administered weekly for ten weeks, with a specific emphasis on comparing parameter changes between DOX + DEX + CVD and DOX + DEX groups. Evaluation included alterations in body weight, feed and water consumption, and analysis of NADPH_2_, NADP^+^, NADPH_2_/NADP^+^, lipid peroxidation, oxidized DNA, and mRNA for superoxide dismutase 2 and catalase expressions in cardiac muscle. The iron management panel included markers for iron, transferrin, and ferritin. Liver abnormalities were assessed through histological examinations, aspartate transaminase, alanine transaminase, and serum albumin level measurements. During weeks 11 and 21, reduced NADPH_2_ levels were observed in almost all examined groups. Co-administration of DEX and CVD negatively affected transferrin levels in DOX-treated rats but did not influence body weight changes. Ascites predominantly resulted from cardiac muscle dysfunction rather than liver-related effects. The study’s findings, exploring the impact of DEX and CVD on DOX-induced cardiotoxicity, indicate a lack of scientific justification for advocating the combined use of these drugs at histological, biochemical, and molecular levels.

## 1. Introduction

Doxorubicin (DOX) is a potent cytostatic agent widely employed in the treatment of various solid tumors, as well as hematologic malignancies [1,2]. Despite its therapeutic efficacy, the utilization of DOX is constrained by its notable adverse effects, particularly its potential to induce life-threatening cardiac toxicity [1,3,4,5]. Additionally, DOX can adversely affect bone marrow function and cause hepatic injury [6,7,8]. The manifestation of DOX-induced cardiotoxicity appears in various forms, with dilated cardiomyopathy being the most severe and potentially life-threatening consequence, often emerging years after the completion of DOX therapy. Unfortunately, this condition is irreversible and lacks pharmacological interventions for treatment [9,10]. The focus of over five decades of research has been to comprehend the pathomechanism governing the progression of dilated cardiomyopathy, with the aim of developing interventions to impede its course. DOX-induced cardiomyopathy is a multifaceted condition with diverse contributing factors. Oxidative stress, inflammation, mitochondrial damage, calcium homeostasis disruption, ferroptosis, autophagy, and apoptosis are among the identified mechanisms implicated in the pathogenesis of this condition [11]. Furthermore, a comprehensive strategy for the development of DOX-induced cardiomyopathy is not feasible.

Dexrazoxane (DEX) is the sole FDA-approved medication for mitigating DOX-related cardiomyopathy [12]. DEX acts by binding to iron, reducing the pool of metal ions available for complex formation with anthracyclines, thereby reducing the generation of superoxide radicals and potentially limiting the production of reactive free radicals in the Fenton and Haber–Weiss reactions [13,14]. DOX disrupts the regular catalytic cycle of topoisomerase 2β (TOP2β), causing DNA strand breaks and potential cardiomyocyte death [10,15]. Recent evidence suggests that DEX prevents DOX from binding to TOP2β, protecting the heart from harmful DOX effects [1,10,15,16]. Despite growing understanding of DEX’s cardioprotective properties, its use is restricted by the FDA to patients with advanced breast cancer, adult patients with soft tissue sarcomas, or small-cell lung cancer [17]. Moreover, DEX administration does not entirely eliminate the risk of DOX-induced cardiac damage [18]. Therefore, exploring alternative strategies for preventing DOX cardiomyopathy is imperative.

Over several decades, numerous compounds, distinct from DEX, have undergone evaluation primarily due to their antioxidant properties. However, only select agents have demonstrated cardioprotective attributes in laboratory experiments, making them suitable for entry into clinical trials, i.e., carvedilol (CVD) (NCT04023110). CVD, a β-blocker endowed with antioxidant properties, has been subject to examination in two separate randomized clinical trials, both of which demonstrated its efficacy in preventing dilated cardiomyopathy induced by DOX [19,20]. Notably, CVD is, as an inhibitor of mitochondrial complex I, a critical contributor to the supply of NADH for the DOX redox cycle. This inhibitory action is assumed to be the mechanistic basis for the development of DOX-induced dilated cardiomyopathy [21,22]. The study conducted by Avila et al. in 2018 demonstrated the superior efficacy of CVD compared to propranolol in preventing DOX-induced cardiomyopathy [21].

In our previous investigation, we showed a decrease in left ventricular function, as manifested by a reduction in left ventricular ejection fraction (LVEF), concomitant with indications of cardiomyocyte necrosis, demonstrated by elevated levels of cardiac troponin I (cTnI) in the systemic circulation. The treatment with DEX effectively ameliorated these perturbations. However, the co-administration of DEX and CVD, aimed at potentiating the protective effect, failed to yield the anticipated outcome. In the present study, our focal point is to elucidate the characteristics associated with oxidative stress, with a specific emphasis on the redox equilibrium within the cardiac muscle. A comprehensive assessment was undertaken to ascertain whether the administration of CVD to rats concurrently subjected to DOX and DEX induced any untoward effects on hepatic physiology. Additionally, based on our antecedent observations of fluid accumulation in the abdominal cavity of rats treated with DOX, our objective was to determine whether this phenomenon was exclusively attributable to cardiac etiology or exhibited a discernible hepatic component.

## 2. Results

The experiment was performed as previously described [23]. Briefly, rats were acclimatized before the study treatment began. Rats were administered with drugs every week for ten weeks. One week after the end of the drug administration, half of the rats per group were euthanized. The remaining rats were euthanized eleven weeks after the end of the drug administration. To illustrate the design of the experiment, we have prepared Figure 1. Details on the methodology and the exact division into groups can be found in Section 4.2.—Experimental Design.

Considering the research objective, the primary focus in describing the results is on the impact of CVD on the evaluated parameters in rats treated with DOX and DEX. Specifically, comparison is made between the DOX + DEX + CVD group and the DOX + DEX group.

### 2.1. Body Mass Dynamics and Consumption Patterns

The acquired data demonstrate the disruptive impact of DOX on the process of body weight progression (Table 1). This is substantiated by a significant reduction in the average body weight across all experimental groups, relative to the control group, in the 11th week of the study. In the DOX + DEX + CVD group, these changes were significant as early as week 8 of the experiment. This observed effect persists into the 21st week of the trial, specifically 11 weeks post the discontinuation of medication in groups administered protective agents in conjunction with DOX. This observed phenomenon appears to be an outcome of the combined influence of both DOX and the protective agents. Including preventive agents had no beneficial impact on the rats’ weight increase (in groups of DOX + DEX, DOX + CVD, DOX + DEX + CVD compared to control). Interestingly, this phenomenon increased in the DOX + DEX group compared to the group of rats given DOX alone (Table 1).

In the weight increment evaluation between weeks 11 and 21 of the study (Δ = T_21_–T_11_; Table 2), aberrations in weight gain were evident across all experimental groups when compared to the control group. Notably, the co-administration of CVD with DOX + DEX did not exert any noticeable influence on weight gain.

The examination of feed and water intake up to the 11th week of the experiment, namely during the period when the rats were administered the investigated chemicals, revealed no impact of CVD on these measures in rats receiving DOX + DEX (DOX + DEX + CVD vs. DOX + DEX group). There was a reduction of c.a. 10% in feed (Figure 2a) and 30% in water (Figure 2b) consumption in all research groups compared to the control group.

### 2.2. Biochemical Analyses

In this study, we investigated the role of oxidative stress in DOX-dependent late cardiomyopathy, as previously demonstrated in our earlier research on the same rats [23]. Specifically, we examined parameters associated with the redox balance, which is crucial for oxidative stress in the heart muscle (Table 3). The levels of NADPH_2_, NADP^+^, and the NADPH_2_/NADP^+^ ratio were evaluated because NADPH_2_ is involved in generating free radicals and regenerating reduced glutathione (GSH), which is a small-molecule antioxidant defense component. No disparities between the DOX + DEX + CVD and DOX + DEX groups were observed in these parameters (Table 3). A decline in NADPH_2_ concentration was seen in nearly all groups analyzed during weeks 11 and 21 of the study. In contrast, the DOX + CVD group exhibited a higher concentration of this parameter throughout the 11th week of the research than the control group. However, no noticeable differences were observed among the examined groups compared to the control, DOX, and DOX + DEX groups. The NADPH_2_/NADP^+^ ratio showed a notable drop in the DOX + DEX + CVD group compared to the control group. This decrease was also detected in the 21st week of the research.

The levels of oxidative damage to both total DNA (oxDNA) and lipids (LPO, ipid peroxidation) were measured. No disparities were detected between the DOX + DEX + CVD and DOX + DEX groups (Table 3). Most study groups observed a rise in oxDNA levels compared to the control group. At week 11 of the study, the presence of DEX and DEX combined with CVD resulted in an increase in the quantity of oxDNA in rats treated with DOX (DOX + DEX vs. DOX and DOX + DEX + CVD vs. DOX groups). In contrast to what was expected, significantly lower levels of LPO vs. control marker were observed in nearly all research groups.

No significant alterations were observed in the measures used to evaluate liver necrosis and function, namely, AST, ALT, and albumin, in both the control group and the groups treated with DOX and DOX + DEX (Table 4).

Serum transferrin level was the sole variable that showed significant differences between the DOX + DEX group and the DOX group, as well as between the DOX + DEX + CVD group and the DOX + DEX group (Table 5). DEX inhibited the rise in transferrin caused by DOX at week 11. Furthermore, it was discovered that the administration of CVD to rats receiving DOX + DEX at week 21 resulted in a notable elevation in transferrin levels. All research groups exhibited a decrease in serum iron content by week 11. By week 21, the decreased concentration is only present in the DOX + DEX + CVD group, whereas in the other groups, it is not significantly different from the control group.

### 2.3. Assessment of Gene Expression

The levels of gene expression for mitochondrial superoxide dismutase (*Sod2*) and catalase (*Cat*), which play a crucial role in the enzymatic antioxidant defense system, were also measured. A noticeable influence on the mRNA expression levels of *Sod2* and *Cat* genes was identified subsequent to drug administration in the 11th week of the experiment (Table 6). Specifically, *Sod2* was downregulated in the DOX and DOX + DEX + CVD groups, while it was overexpressed in the DOX + DEX and DOX + CVD groups relative to the control group. The *Cat* gene was observed to be overexpressed across all experimental groups. Notably, alterations in expression patterns during the 21st week of the study did not show statistical significance.

### 2.4. Histological Staining

The microscopic image showed alterations characterized by glycogen accumulation, an augmentation in the number of inflammatory cells, and the emergence of individual apoptotic cells in rats treated with CVD in combination with DOX + DEX (group DOX + DEX + CVD vs. DOX + DEX) (Figure 3a,b and Figure 4a–f, Table 7). Groups that received DOX alone exhibited intensification or appearance of all evaluated morphological characteristics compared to the control group. DEX effectively inhibited all DOX-induced alterations (DOX + DEX vs. DOX). Conversely, the administration of CVD to rats treated with DOX had a negligible impact on the observed alterations in the DOX group.

## 3. Discussion

This study represents a continuation of the findings from previous research [23]. It emphasizes the complex interplay of redox balance, iron metabolism, and drug interactions in the context of anthracycline-induced cardiotoxicity. It provides valuable insights into potential treatment strategies and highlights the need for cautious consideration of drug combinations in addressing cardiotoxic effects associated with DOX administration. Additionally, considering the previously observed DOX-induced ascites, this study explores the potential involvement of the liver in ascites development.

The interpretation of NADPH_2_ is challenging, given its seemingly contradictory roles. On one hand, it produces free radicals when interacting with compounds possessing a quinone structure [4,24]. On the other hand, it assumes a pivotal role in the regeneration of glutathione reductase, a key enzyme responsible for scavenging free radicals by facilitating the production of reduced glutathione. Noteworthy enzymes, including cytochrome P-450 reductase (EC 1.6.2.4) [25] and NO synthase (EC 1.14.13.39) [26,27], utilize NADPH_2_ as a cofactor to transfer an electron from this nucleotide to anthracyclines. The anthracycline radical surrenders an electron to molecular oxygen, giving rise to a superoxide anion radical [4]. A potential concern in this reaction lies in the anthracycline molecule functioning as a catalyst rather than undergoing consumption in a typical substrate–product response. Consequently, this process may persist until the depletion of NADPH_2_ stores or the elimination of anthracycline from the cell due to metabolic alterations.

A substantial concern arising from the analysis of NADPH_2_ outcomes pertains to the etiology and mechanism underpinning the consistently reduced concentration of this nucleotide. It becomes apparent that the free radical theory postulates that late-stage heart failure’s occurrence is attributable to the lingering free radical damage arising from the presence of DOX in the body. Consequently, the harmful effects induced by DOX lead to the generation of reactive oxygen species (ROS), exacerbating damage and triggering further ROS synthesis. The escalating production of ROS necessitates a corresponding increase in the involvement of NADPH_2_-dependent free radical scavengers. The observation that the diminished NADPH_2_ level persists for 11 weeks following the cessation of compound administration (at week 21) provides evidence that the body has engaged its adaptive mechanisms but remains incapable of restoring NADPH levels to those observed in the control group. Consequently, this interpretation aligns with the assumptions of the free radical theory. However, the equilibrium of the NADPH_2_/NADP^+^ redox buffer is maintained in all groups, with the exception of the DOX + DEX + CVD group, which shows no significant deviation from the control group. This finding underscores the detrimental impact of CVD when administered concomitantly with DOX and DEX.

It is noteworthy to observe that the administration of DEX in rats subjected to DOX and DOX + CVD treatment results in an elevation in oxidative DNA damage, manifested by heightened levels of oxDNA in both the DOX + DEX and DOX + DEX + CVD groups, in contrast to the absence of such alterations in the DOX + CVD group. Unconventional alterations in LPO were identified in our study, deviating from the anticipated findings of elevated levels of LPO products observed in previous short-term investigations involving DOX [28,29,30,31]. However, our research revealed a decrease in LPO across nearly all experimental groups. Contrary to expectations, the combination of CVD and DEX did not exhibit advantageous synergy in any of the assessed biochemical redox balance parameters.

Concerning *Sod2* mRNA expression level, the observed outcome was considered unfavorable due to a notable decrease in expression levels within the DOX + DEX + CVD group compared to the control group. Conversely, in the case of *Cat* mRNA expression level, observed overexpression in the DOX + DEX group was restored to control levels. However, the interpretive context of this observed impact remains uncertain, as the elevated value may be attributed to the adaptation of cardiac muscle cells to the combined influence of DOX and DEX.

The investigation of characteristics associated with iron metabolism becomes imperative due to its substantial influence on the cardiotoxicity of anthracyclines, given its involvement in free radical reactions. The formation of a complex between iron and DOX results in the production of ROS at a significantly enhanced efficiency compared to DOX alone [32]. Substantiating the role of iron in the cytotoxicity induced by DOX is the observed protective effect when employing iron chelators. DEX, one of the chelators, facilitates the extraction of iron from mitochondria that have accrued an excessive amount of this element due to the effects of DOX [33]. The elevation of iron concentration can instigate the Fenton and Haber–Weiss reactions, elucidating the association between excessive iron in cardiomyocyte mitochondria and oxidative stress [34]. Additionally, iron assumes a pivotal role in sustaining redox balance and is implicated in catalase activity.

This investigation analyzes the concentration of iron in blood serum, specifically focusing on transferrin and ferritin. In the plasma, iron exists as a soluble compound bound to transferrin. This process entails the binding of iron being transported with the transferrin receptor, forming a complex known as transferrin–dimeric transferrin receptor (TfR). Facilitating the passage of iron across the cell membrane through endocytosis, this complex plays a crucial role in cellular iron transport. Ferrous ions (Fe^2+^) gain entry into the cytosol via divalent metal transporter 1 (DMT1). Upon cellular entry, iron integrates into the labile iron pool, which is available for utilization in essential cellular enzymes such as heme and iron–sulfur clusters, or it may be stored by the iron storage protein ferritin [33,35].

Diminished iron concentration may arise from either intestinal malabsorption or the accumulation of iron in tissues. The propensity for iron accumulation is pervasive in various bodily tissues, with the liver exhibiting remarkable susceptibility to such accumulation, often leading to the prevalent manifestation of cirrhosis and multifocal primary liver cancer. The occurrence of diabetes is linked to pancreatic damage, whereas heart failure and frequent arrhythmias are the result of damage to the heart muscle [36]. However, this occurs when excessive iron is absorbed from the gastrointestinal tract.

In light of the simultaneous elevation in iron levels across all groups, a critical inquiry arises concerning the potential occurrence of organ deposition even in the context of diminished serum iron levels. The observed augmentation in transferrin levels accompanied by a concurrent reduction in iron levels is likely indicative of compensatory or adaptive mechanisms aimed at preserving a stable cellular iron concentration.

Observed disturbances in weight growth were noted between the study groups and those subjected to DOX or DOX + DEX. Consequently, the induction of CVD in rats undergoing DOX + DEX treatment does not exert an influence on modifications in these parameters.

Our prior investigation [23] revealed that the administration of DOX alone induces ascites characterized by the presence of translucent fluid devoid of blood. By the 21st week, the mortality rate within the DOX group increased by 50%, and the ascites transitioned to a state characterized by the presence of bloody purulent fluid. Simultaneous administration of DEX or DEX + CVD effectively prevented mortality and the development of ascites caused by DOX. However, the administration of CVD alone failed to confer protection against these outcomes. Consequently, within the DOX + DEX + CVD group, the protective effect was solely attributed to DEX. In the exploration of protective mechanisms, it is evident that blocking β1 and β2 adrenergic receptors, along with the antioxidant effect associated with CVD, does not preclude ascites and may even contribute to its development by diminishing stroke and output capacity. The essential focus for mitigating mortality and ascites development lies in understanding the impact of DEX on iron metabolism and its influence on TOPIIβ.

Ascites is primarily linked to heart failure and liver dysfunction. The administration of DOX intraperitoneally may impact the occurrence of ascites: peritonitis resulting from the presence of DOX in the cavity of the peritoneum. It is crucial to take this into account. Our previous study [23] revealed substantial associations between alterations in cardiac ejection fraction and the development of ascites across different groups. Hence, the occurrence of heart failure during the 21st week and its mitigation by DEX may be predominantly reliant on heart failure itself. In this study, we aimed to assess the influence of liver lesions on mortality rates and ascites development. Morphological evaluations demonstrate that DEX exerts a protective effect against pathological alterations induced by DOX, distinct from the characteristic features of CVD. The inclusion of CVD (DOX + DEX + CVD group) revealed a detrimental impact compared to the DOX + DEX group. In summary, DOX induced morphological alterations that were inhibited by DEX.

However, it is evident that minor alterations, such as the presence of immune cell clusters, cell death, and tissue decay, do not exert a discernible influence on individual survival rates or the occurrence of abdominal fluid accumulation. The absence of fluctuations in the levels of serum transaminases—specifically AST and ALT—in comparison to the control group underscores the limited relevance of individual instances of necrosis observed microscopically to liver functionality. The assessment of albumin production capacity may serve as an additional indicator of liver function in the context of ascites. Nevertheless, no significant differences were noted in the levels of serum albumin concentration across any of the study groups when compared to the control group or groups subjected to DOX and DOX + DEX treatments. Consequently, it can be concluded that the observed ascites primarily stem from a mechanism associated with heart failure.

## 4. Materials and Methods

### 4.1. Animals

A total of one hundred male Wistar rats, aged eight weeks, were obtained from the Experimental Medicine Center at the Medical University of Lublin, Poland. During the experiment, the rats were kept in carefully regulated environmental conditions, with a temperature range of 22 ± 3 °C, relative humidity maintained at 50 ± 5%, and a consistent 12-h light/dark cycle. The animals had unrestricted access to potable water and a standardized rodent meal. Body weight, feed, and water consumption were recorded weekly. The procedures were executed between the hours of 9 a.m. and 3 p.m. The Local Ethical Committee approved the animal study protocol (123/2018) for Animal Experiments at the University of Life Sciences in Lublin, Poland (approved on 3 December 2018). The experimental animal protocols complied with the European Committee Directive for Care and Use of Laboratory Animals (2010/63/EU). The animals were under constant veterinarian monitoring, and all possible measures were taken to minimize any injury.

### 4.2. Experimental Design

Before the trial, the rats were subjected to a 7-day acclimatization phase. Afterward, the animals were randomly assigned to five study groups. Initially, there were twenty rats in each experimental group. Half of the animals were euthanized in the 11th week, which was one week after the completion of the treatment period. The remaining animals were euthanized in the 21st week, ten weeks after the administration concluded.

The experimental groups were as follows: control group (CTR); DOX without any prior treatment (DOX); DEX and CVD pretreatment 30 min before DOX administration (DOX + DEX + CVD); DEX pretreatment 30 min before DOX administration (DOX + DEX); CVD pretreatment 30 min before DOX administration (DOX + CVD) (Table 8). Throughout the experiment, there was a fluctuation in the number of animals in the two groups (DOX in the 21st week of study, n = 5; DOX + CVD in the 21st week of study n = 6).

DOX, DEX, and CVD were obtained from Merck in Darmstadt, Germany. The solutions were produced in a volume of 0.01 mL per gram of body weight right before being administered. The rats were euthanized with 3.5% isoflurane anesthesia and decapitation. Subsequently, their hearts and livers were collected for pathology and molecular research. After collecting blood, it was centrifugated to obtain serum for later biochemical examination.

### 4.3. Biochemical Analysis

NADP+ and NADPH_2_ concentrations were measured in 20 mg of collected tissue using NADP/NADPH Assay Kit (BioChain, San Francisco Bay Area, CA, USA) according to the manufacturer’s protocol. An absorbance was measured at λ = 565 nm at T0 and T15. The analysis was performed in three technical repetitions.

The lipid peroxidation assay relies on measuring the levels of malondialdehyde and 4-hydroxyalkenals (MDA  +  4HAE) (OxisResearch, N Cutter Circle, Portland, USA). The assessment is based on the reaction between a chromogenic reagent R1 (N-methyl-2-phenylindole) and malondialdehyde (MDA) and 4-hydroxyalkenals (4HAE) at a temperature of 45 °C. The reaction between two molecules of R1 and one molecule of MDA or 4-hydroxyalkenals results in the formation of a chromophore that exhibits maximum absorbance at 586 nm. The quantity of MDA, along with 4-hydroxyalkenals, in methane sulfonic acid was measured to indicate lipid peroxidation.

AST, ALT, albumin, iron, transferrin, and ferritin were analyzed in the rats’ blood serum using a Sandwich enzyme immunoassay ELISA Kit (Cloud-Clone Corp., Houston, TX, USA) according to the manufacturer’s instructions. Color changes were measured spectrophotometrically using a PowerWave microplate spectrophotometer (BioTek Instruments, Winooski, Vermont, MA, USA) at a wavelength λ = 450 nm, and measured concentrations were determined in relation to standard curves.

The oxidative DNA damage was assessed by measuring the number of abasic sites (AP sites). ROS create AP sites, which cause significant DNA damage. The DNA was extracted using the Syngen DNA Mini Kit (Syngen, Poland) following the directions provided by the manufacturer. The genomic DNA concentration was determined using the MaestroNano Micro-Volume Spectrophotometer (Maestrogen Inc., Hsinchu, Taiwan) and then adjusted to a 100 ng/µL concentration in the TE buffer. The quantity of AP sites was assessed using the DNA Damage Quantification Kit (Dojindo, Kumamoto, Japan) following the instructions provided by the manufacturer. The technique relies on the selective interaction between an aldehyde-reactive probe (ARP; N′-aminooxymethylcarbonylhydrazin-D-biotin) and an aldehyde group found on the open-ring form of AP sites. AP sites were labeled with biotin residues and measured using an avidin–biotin assay. The quantification was performed by detecting the peroxidase-conjugated avidin at 650 nm using a PowerWave microplate spectrophotometer (BioTek Instruments, Winooski, Vermont, MA, USA).

### 4.4. Molecular Studies (qPCR Analysis)

The investigations involving the utilization of rat heart mRNA were made as described previously [23]. Total RNA was extracted from 50 mg of left ventricular tissue using the TRIzol reagent (Invitrogen, Carlsbad, CA, USA). The MaestroNano NanoDrop spectrophotometer (Maestrogen, Hsinchu, Taiwan) was used to assess the concentration and purity of the isolated RNA. Only high-quality RNA with an A260/280 ratio of 1.8–2.0 was chosen for further investigations. Afterward, cDNA synthesis was conducted using a cDNA reverse transcription kit (Applied Biosystems, Foster City, CA, USA) with the following parameters: 25 °C for 10 min, 37 °C for 120 min, and finally 85 °C for 5 min.

The relative expression of the investigated genes was quantitatively assessed using real-time PCR. This was done by utilizing the high-throughput SmartChip MyDesign Chip system from WaferGen Bio-Systems in Fremont, California, CA, USA, along with the PowerUp SYBR Green Master Mix from Applied Biosystems in Foster, CA, USA. Each reaction was repeated four times. The reaction profile consisted of heating at 95 °C for 2 min, followed by 45 cycles of heating at 95 °C for 15 s, 57 °C for 15 s, and 72 °C for 1 min. A temperature ramp of 0.4 °C/s was used to construct a melt curve, which reached a maximum temperature of 97 °C.

To confirm the accuracy of the data, a thorough examination of amplification, Tm, and Ct values was performed, eliminating any data points that deviated significantly from the norm, before calculating ΔΔCt and evaluating the magnitude of change in mRNA levels. The *Rpl32* and *Tbp* housekeeping genes were utilized for data standardization. The results were reported as the mean relative quantification (RQ) using the formula RQ = (2 ^−ΔΔCt^).

Detailed information on the genes and primer sequences used in the study is provided in Table 9.

### 4.5. Histological Staining

Pieces of the liver were procured from each individual and preserved in buffered 10% formalin (pH = 7.4). Subsequently, these specimens were subjected to processing into paraffin blocks. Slides of four micrometers were sliced using a microtome and underwent staining with hematoxylin and eosin, as well as Periodic Acid–Schiff (PAS) to facilitate the visualization of liver alterations induced by the experimental treatments. A proficient blinded pathologist conducted the assessment of the slides using a light microscope. From one liver, sections of two different lobes for each staining were evaluated in each animal (20 slides of hematoxylin and eosin per group; 20 for PAS per group). The categorization of histological changes was designated as follows: “-”, denoting no observed changes; “+”, indicating changes of minor intensity; “++”, representing moderate changes; and “+++”, changes of major intensity.

### 4.6. Statistical Analysis

Statistical analysis was performed using GraphPad Prism version 6.04 for Windows, GraphPad Software, www.graphpad.com (accessed on 29 December 2023, and 5 January 2024). The normal distribution of the variables was verified using the Shapiro–Wilk test. Tukey’s post hoc tests (HSD and Spjotvolla–Stoline) were applied in combination with a one-way ANOVA for the statistical analysis. The changes in body weight of rats over 21 weeks of the study were analyzed using one-way ANOVA for repeated measurements and Tukey’s HSD test. The incidence of histological abnormalities was evaluated by chi2 test. The data were calculated as mean ± SD. When *p*-value was under 0.05, differences among the groups were regarded as statistically significant.

## 5. Conclusions

Based on the NADPH_2_ concentration, a significant decrease in the reduction potential in the myocardium was demonstrated in almost all study groups, both at weeks 11 and 21 of the study. Contrary to expectations, CVD did not demonstrate beneficial synergy with DEX in relation to any of the biochemical redox balance parameters tested in DOX-treated rats. A significant effect of CVD on the hearts of rats receiving DOX + DEX simultaneously was observed on the mRNA levels of *Sod2* and *Cat*. In the case of *Sod2*, it had an unfavorable effect, and in the case of *Cat*, it is difficult to determine whether the change was beneficial or not at this stage. Administration of CVD to rats receiving DOX + DEX resulted in an unfavorable interaction in terms of serum transferrin levels (at week 21), as this parameter increased approximately two times compared to the DOX + DEX group. The results of this study at the functional, biochemical, and molecular levels confirm that there is no basis for recommending the simultaneous administration of DEX and CRV in DOX-induced cardiotoxicity.

## Figures and Tables

**Figure 1 ijms-25-02219-f001:**
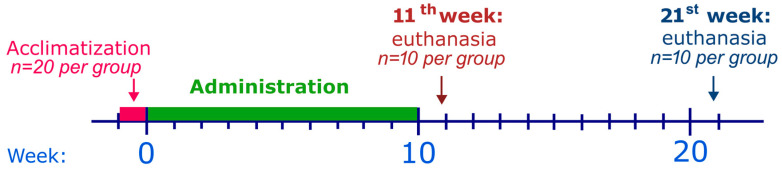
The study scheme. CVD, carvedilol; DOX, doxorubicin; DEX, dexrazoxane.

**Figure 2 ijms-25-02219-f002:**
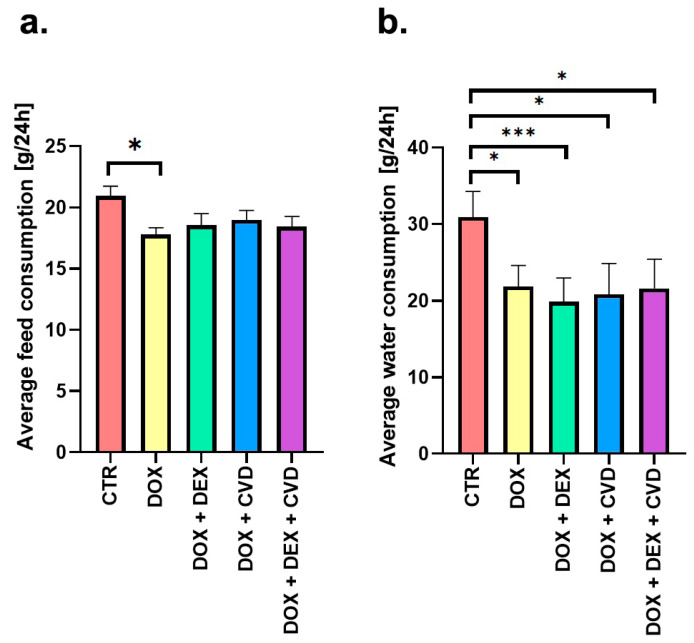
Average feed (**a**) and water (**b**) consumption per rat per day during the drug administration period. The data are presented as mean ± standard deviation of the gain in body mass [g]. Statistical significance: * *p* ≤ 0.05, *** *p* ≤ 0.001 vs. control group (one-way ANOVA with Tukey’s post hoc test). CTR, control; CVD, carvedilol; DOX, doxorubicin; DEX, dexrazoxane.

**Figure 3 ijms-25-02219-f003:**
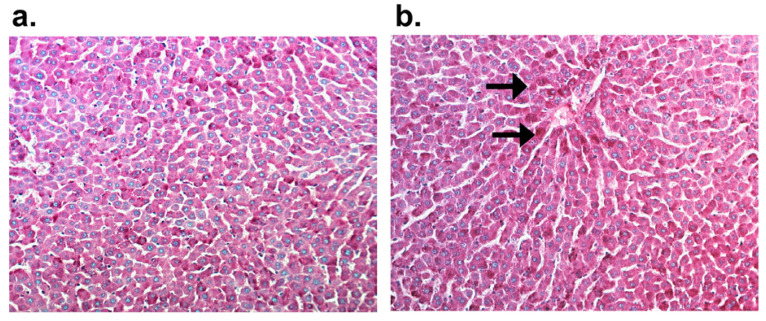
Periodic Acid–Schiff staining, mag.200×. (**a**) Control group; (**b**) Glycogen accumulation in the DOX group hepatocytes (→; the 11th week of the study.

**Figure 4 ijms-25-02219-f004:**
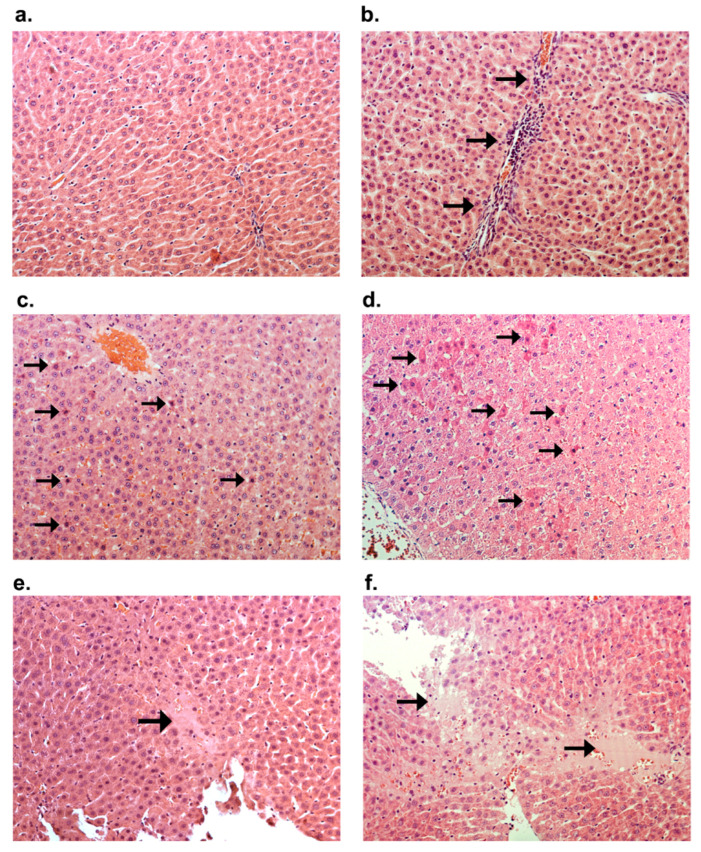
Hematoxylin and eosin staining, mag. 200×. (**a**) Control group; (**b**) Local inflammatory cell accumulation in the DOX + DEX + CVD group hepatocytes (→), the 21st week of the study; (**c**) Scattered single apoptotic cells (→) in the liver of rat treated with DOX; the 11th week of the study; (**d**) Scattered single apoptotic cells (→) and single mononuclear (inflammatory) cells in the liver of rat treated with DOX + DEX + CVD in the 21st week of the study; (**e**) Focal necrosis of hepatocytes in the liver of rat treated with DOX; the 11th week of the study; (**f**) Focal necrosis of hepatocytes in the liver of the DOX + CVD group; the 11th week of the study.

**Table 1 ijms-25-02219-t001:** The effect of administered drugs on changes in body weight of rats over 21 weeks of the study. Data are presented as mean ± standard deviation of body mass [g]. Statistical significance: * *p* ≤ 0.05 vs. control group; # *p* ≤ 0.05 vs. DOX (one-way ANOVA with Tukey’s post hoc test).

Week of Study	Study Group				
	CTR	DOX	DOX + DEX	DOX + DEX + CVD	DOX + CVD
0	208.00 ± 12.31	206.60 ± 11.12	205.50 ± 11.91	204.66 ± 10.77	205.70 ± 22.67
1	234.80 ± 13.52	237.40 ± 16.58	230.80 ± 11.57	229.80 ± 13.56	230.20 ± 28.10
2	273.40 ± 18.99	271.60 ± 20.27	262.10 ± 18.62	258.20 ± 17.51	261.20 ± 33.81
3	297.10 ± 23.56	290.80 ± 20.61	275.70 ± 11.40	274.90 ± 20.24	277.20 ± 28.82
4	313.70 ± 27.20	299.80 ± 25.20	284.30 ± 22.54	286.30 ± 21.70	301.90 ± 28.21
5	331.70 ± 26.18	320.50 ± 33.68	301.20 ± 18.88	295.60 ± 20.50	305.00 ± 35.96
6	342.90 ± 28.63	316.40 ± 26.31	306.55 ± 14.73	297.20 ± 20.00	319.60 ± 27.90
7	355.40 ± 29.01	336.30 ± 33.70	312.10 ± 20.02	299.90 ± 21.71	337.30 ± 24.98
8	362.80 ± 28.11	326.50 ± 25.09	316.55 ± 21.70	311.80 ± 21.76 *	330.30 ± 29.48
9	364.60 ± 37.81	331.10 ± 24.11	322.60 ± 20.31	320.90 ± 20.94 *	338.70 ± 30.27
10	378.90 ± 31.13	331.00 ± 17.18	321.20 ± 17.93	317.70 ± 21.63 *	334.33 ± 30.46
11	394.60 ± 33.41	331.20 ± 13.61 *	324.30 ± 16.94 *	323.60 ± 23.66 *	336.00 ± 19.42 *
21	540.50 ± 42.21	477.10 ± 24.23 *	418.90 ± 22.56 *^,#^	439.50 ± 28.74 *	444.30 ± 27.32 *

CTR, control; CVD, carvedilol; DOX, doxorubicin; DEX, dexrazoxane.

**Table 2 ijms-25-02219-t002:** The differences in body weight during and after administration of the test substances. The data are presented as a delta mean ± standard deviation of the gain in body mass [g]. Statistical significance: * *p* ≤ 0.05 vs. control group (one-way ANOVA with Tukey’s post hoc test).

The Differences in Body Weight	Study Group				
	CTR	DOX	DOX + DEX	DOX + DEX + CVD	DOX + CVD
Δ = T_11_–T_0_	186.6 ± 33.8	127.0 ± 13.49 *	118.6 ± 10.7 *	117.3 ± 21.36 *	126.7 ± 14.1 *
Δ = T_21_–T_11_	145.9 ± 10.7	108.6 ± 7.75 *	94.6 ± 6.89 *	115.9 ± 12.84 *	108.3 ± 11.45 *

CTR, control; CVD, carvedilol; DOX, doxorubicin; DEX, dexrazoxane.

**Table 3 ijms-25-02219-t003:** Statistical differences between biochemical parameter levels in rats’ hearts sacrificed in the 11th or 21st week of study. The values are presented as a mean ± SD. Statistical significance: * *p* ≤ 0.05 vs. control group; # *p* ≤ 0.05 vs. DOX (one-way ANOVA with Tukey’s post hoc test).

Biochemical Parameter	Week of the Study	Study Group				
		CTR	DOX	DOX + DEX	DOX + DEX + CVD	DOX + CVD
NADPH_2_ [µM]	11	1.00 ± 0.19	0.51 ± 0.12 *	0.59 ± 0.11 *	0.69 ± 0.15 *	1.55 ± 0.73 ^#^
	21	1.33 ± 0.37	0.83 ± 0.17 *	0.55 ± 0.22 *	0.29 ± 0.13 *	0.30 ± 0.29 *^,#^
NADP^+^ [µM]	11	0.51 ± 0.12	0.47 ± 0.16	0.39 ± 0.19	0.60 ± 0.17	0.62 ± 0.44
	21	0.96 ± 0.42	0.80 ± 0.27	0.82 ± 0.43	0.42 ± 0.48	0.55 ± 0.28
Ratio of NADPH_2_/NADP^+^	11	1.72 ± 0.34	1.02 ± 0.40	2.30 ± 2.17	1.05 ± 0.33 *	1.86 ± 1.20
	21	1.29 ± 0.32	1.03 ± 0.45	0.89 ± 0.72	0.69 ± 0.33 *	0.70 ± 0.33
oxDNA [number/100kbp]	11	0.50 ± 0.16	1.02 ± 0.49	2.66 ± 0.84 *^,#^	3.42 ± 1.60 *^,#^	2.30 ± 0.86 *
	21	0.62 ± 0.13	1.76 ± 0.69 *	1.32 ± 0.79	1.18 ± 0.32 *	1.54 ± 0.52 *
LPO [µM MDA]	11	7.54 ± 1.91	1.18 ± 0.66 *	0.96 ± 0.32 *	1.18 ± 0.66 *	0.63 ± 0.31 *
	21	7.26 ± 1.23	3.77 ± 1.91 *	2.22 ± 1.61 *	4.92 ± 2.88	2.50 ± 1.94 *

CTR, control; CVD, carvedilol; DOX, doxorubicin; DEX, dexrazoxane.

**Table 4 ijms-25-02219-t004:** Biochemical parameter levels in rats’ serum in the 11th or 21st week of study. The values are presented as a mean ± SD.

Biochemical Parameter	Time	Study Group				
		CTR	DOX	DOX + DEX	DOX + DEX + CVD	DOX + CVD
AST [IU/L]	11	63.24 ± 14.85	77.96 ± 21.90	60.09 ± 10.14	59.52 ± 9.38	68.74 ± 20.03
	21	51.75 ± 6.32	63.40 ± 14.77	49.82 ± 10.38	51.35 ± 14.01	42.22 ± 11.63
ALT [IU/L]	11	15.97 ± 2.19	21.78 ± 3.21	17.98 ± 2.09	17.94 ± 1.99	21.08 ± 3.26
	21	15.27 ± 1.27	19.29 ± 4.44	16.59 ± 1.93	16.37 ± 3.58	20.73 ± 7.62
Albumin [g/dL]	11	3.27 ± 0.37	3.47 ± 0.17	3.92 ± 0.46	3.70 ± 0.26	3.93 ± 0.31
	21	3.36 ± 0.48	3.12 ± 0.38	3.11 ± 0.62	3.00 ± 0.40	3.24 ± 0.91

CTR, control; CVD, carvedilol; DOX, doxorubicin; DEX, dexrazoxane.

**Table 5 ijms-25-02219-t005:** Study treatment impact on biochemical parameters associated with iron metabolism. Statistical significance: * *p* ≤ 0.05 vs. control group; # *p* ≤ 0.05 vs. DOX; † *p* ≤ 0.05 vs. DOX + DEX (one-way ANOVA with Tukey’s post hoc test).

Biochemical Parameter	Time	Study Group				
		CTR	DOX	DOX + DEX	DOX + DEX + CVD	DOX + CVD
Iron (Fe) [µg/dL]	11	124.75 ± 10.11	105.16 ± 8.73 *	96.04 ± 12.65 *	95.54 ± 11.18 *	99.109 ± 12.91 *
	21	121.75 ± 9.42	106.30 ± 15.50	103.55 ± 17.89	96.84 ± 9.51 *	101.84 ± 32.76
Transferrin [mg/dL]	11	424 ± 260	1049 ± 142 *	898 ± 228	1059 ± 183*	1033 ± 228 *
	21	516 ± 326	559 ± 209	507 ± 329	1040 ± 219 *^,#,†^	636 ± 149
Ferritin [µg/L]	11	3482 ± 1068	3740 ± 676	3918 ± 1325	3733 ± 1853	4317 ± 1426
	21	3112 ± 931	2607 ± 550	2736 ± 1217	2380 ± 670	1732 ± 641

CTR, control; CVD, carvedilol; DOX, doxorubicin; DEX, dexrazoxane.

**Table 6 ijms-25-02219-t006:** Results of mRNA expression of *Sod2 and Cat* genes in rats’ hearts euthanized in the 11th or 21st week of study. Data are presented as mean ± standard deviation of relative quantity of mRNA level. Statistical significance: * *p*≤ 0.05 vs. control group; # *p* ≤ 0.05 vs. DOX; † *p*≤ 0.05, DOX + DEX + CVD vs. DOX + DEX group (one-way ANOVA with Tukey’s post hoc test).

Gene	Week of the Study	Study Group				
		CTR	DOX	DOX + DEX	DOX + DEX + CVD	DOX + CVD
*Sod2*	11	1.01 ± 0.18	0.67 ± 0.11 *	1.92 ± 0.25 *^,#^	0.71 ± 0.08 *^,†^	1.52 ± 0.20 *^,#^
	21	1.02 ± 0.22	0.85 ± 0.10	0.92 ± 0.12	0.84 ± 0.12	1.32 ± 0.17 ^#^
*Cat*	11	1.05 ± 0.25	6.77 ± 1.86 *	32.29 ± 8.72 *^,#^	8.02 ± 2.34 *^,†^	21.96 ± 5.76 *^,#^
	21	1.01 ± 0.29	1.43 ± 0.38	1.07 ± 0.22	1.39 ± 0.43	1.67 ± 0.45

CTR, control; CVD, carvedilol; DOX, doxorubicin; DEX, dexrazoxane.

**Table 7 ijms-25-02219-t007:** The presence and intensity of morphological changes in rats’ liver after the study treatment.

Morphological Feature	Week of the Study	Study Group				
		CTR	DOX	DOX + DEX	DOX + DEX + CVD	DOX + CVD
Glycogen accumulation	11	+ (7)	+++ (10)	-	++(7)	+ (8)
	21	+ (7)	-	-	+ (7)	+ (7)
Foci of inflammatory cells	11	-	+ (9)	-	+ (6)	+ (8)
	21	-	+ (10)	-	++ (8)	+ (9)
Single cell death (apoptosis)	11	-	+ (6)	-	-	-
	21	-	-	-	+ (5)	-
Necrosis	11	-	+ (6)	-	-	++ (7)
	21	-	-	-	-	-

-, no changes; +, changes of minor intensity; ++, moderate changes; +++, changes of major intensity, incidence in the group was given in brackets (n = 10). CTR, control; CVD, carvedilol; DOX, doxorubicin; DEX, dexrazoxane.

**Table 8 ijms-25-02219-t008:** The experimental administration design.

Symbol of Group	Type of Group	Administration
CTR	Control (*n* = 20)	0.01 mL 0.9% NaCl per g body weight IP administration once a week for 10 weeks;
DOX	Experimental (*n* = 20)	1.6 mg DOX per kg of body weight IP administration once a week for 10 weeks;
DOX + DEX + CVD	Experimental (*n* = 20)	1 mg CVD per kg of body weight IP administration 30 min prior DOX; 25 mg DEX per kg of body weight IP administration 30 min prior DOX; 1.6 mg DOX per kg of body weight IP administration once a week for 10 weeks;
DOX + DEX	Experimental (*n* = 20)	1.6 mg DOX per kg of body weight IP administration once a week for 10 weeks; 25 mg DEX per kg of body weight IP administration 30 min prior DOX;
DOX + CVD	Experimental (*n* = 20)	1.6 mg DOX per kg of body weight IP administration once a week for 10 weeks; 1 mg CVD per kg of body weight IP administration 30 min prior DOX.

CTR, control; CVD, carvedilol; DEX, dexrazoxane; DOX, doxorubicin; IP, intraperitoneal.

**Table 9 ijms-25-02219-t009:** The symbols and names of the genes, GenBank reference sequence accession numbers, and assay IDs.

Gene Name	Gene Symbol	Primer Sequence (5′ → 3′)		NCBI Reference Sequence
		Left	Right	
Catalase	*Cat*	ACA TGG TCT GGG ACT TCT GG	CAA GTT TTT GAT GCC CTG GT	NM_012520.2
Superoxide dismutase 2	*Sod2*	CAC TGT GGC TGA GCT GTT GT	TCC AAG CAA TTC AAG CCT CT	NM_017051.2
Ribosomal Protein L32	*Rpl32*	AGA TTC AAG GGC CAG ATC CT	CGA TGG CTT TTC GGT TCT TA	NM_013226
TATA box binding protein	*Tbp*	CCT CTG AGA GCT CTG GGA TTG TA	GCC AAG ATT CAC GGT GGA TAC A	NM_001004198.1

Początek formularza.

## Data Availability

The data presented in this study are available on request from the corresponding author.

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
