# Peer review of "Assessment of the Impact of Carvedilol Administered Together with Dexrazoxan and Doxorubicin on Liver Structure and Function, Iron Metabolism, and Myocardial Redox System in Rats"

_ijms, 2024, doi:10.3390/ijms25042219_

Round 1

Reviewer 1 Report

Comments and Suggestions for Authors

The manuscript brings a relevant topic and shows some interesting results regarding cardiac and hepatic toxicity induced by doxorubicin. Despite unexpected, the treatment with dexrazoxane and carvedilol did not prevent these side effects induced by doxorubicin. There are some considerations and suggestions below:

1) I believe that include some figures (such as for the weight gain… Maybe a figure for the study design and doses instead table 8…) could increase the visual interesting for the manuscript.

2) I cannot understand the following statement: “Interestingly, the incidence of this phenomenon was decreased in these groups compared to the group of rats treated alone with DOX.” (Page 3, line 131-132). Please, remove or rewrite the sentence.

3) In discussion, the following sentence was not clear: “Notably, pancreatic damage linked to diabetes, as well as heart failure and recurrent arrhythmias attributable to cardiac muscle impairment, are sequelae of this phenomenon [35]. However, this scenario materializes when an excessive influx of iron occurs from the gastrointestinal tract.”  (Page 10, line 320-323). Please, rewrite.

4) I am not agreeing with this sentence in the discussion: “The observed disturbances in weight growth within the groups administered with 329 DOX are attributed to metabolic disruptions rather than a reduction in the intake of food or water.” First, the authors observed a significantly decrease in water intake among groups that received doxorubicin and in food intake in DOX group compared to C group. Second, there are several studies showing that Dox decrease food intake. Third, the authors did not evaluate any metabolic parameter. The discussion at this point was too speculative.

5) Although ascites is commonly related to cardiac or liver damage, in experimental context, when doxorubicin was administered intraperitoneally, other important factor could influence the incidence of ascites: the peritonitis induced by the presence of doxorubicin in the peritoneal cavity. It is important to take into in mind. Additionally, in the present study, the authors did not show results for ascites. I believe it could be removed from the discussion.

6) Conclusion is too long and only summarizing the main results. Please, rewrite.

7) Design study described 10 experimental groups, but the authors presented results only for 5 groups…

8) Please, correct the RT-PCR acronym. For the real time PCR, the acronym should be qPCR (page 14).

9) Did all the variables have a normal distribution? How were tested the normality? And for histological parameters, how was performed the statistical analysis?

10) The expected results were only observed in histological parameters, which depends on the examinator. It is extremely important detailed the analysis. Is the proficient pathologist blind regarding the groups? How many slides were evaluated? How many histological fields were evaluated?

11) Please, update the references.

12) The tittle should not contain “systemic toxicity”. There is no parameter evaluated regarding systemic toxicity, in my point of view.

13) The last consideration is related to controverse results related to oxidative stress. We know how difficult is published negative results, mainly among experimental studies. For this reason, we try hardly explaining our results based in other positive results present in scientific literature. The mechanisms involved in doxorubicin toxicity are multiples and could act differently between models (acute, subacute, chronic etc.) and moments of side effects onset. Despite it could be difficult to explain the oxidative stress, I think we only need to take on the results.

Author Response

Point-by-point response to the reviewer comments

We have read and carefully considered all the comments. We would like to thank the Reviewer for the constructive suggestions and we hope that our responses meet their expectations.

In the corrected version, we have highlighted (with ’tracked changes’) new sentences, rewritten parts and some other changes. Some of these changes were made because of the comments of another reviewer.

Below we provide our point-by-point response to the Reviewer comments and concerns.

Reviewer 1:

The manuscript brings a relevant topic and shows some interesting results regarding cardiac and hepatic toxicity induced by doxorubicin. Despite unexpected, the treatment with dexrazoxane and carvedilol did not prevent these side effects induced by doxorubicin. There are some considerations and suggestions below:

1) I believe that include some figures (such as for the weight gain… Maybe a figure for the study design and doses instead table 8…) could increase the visual interesting for the manuscript.

Response 1: We agree with the Reviewer’s comment. The study was prepared using various data presentation strategies, including figures, pictures, and tables. We attempted to convert the tables into figures, as suggested by the reviewer. However, we believe that this did not improve the clarity and readability of the results.

2) I cannot understand the following statement: “Interestingly, the incidence of this phenomenon was decreased in these groups compared to the group of rats treated alone with DOX.” (Page 3, line 131-132). Please, remove or rewrite the sentence.

Response 2: We agree with the Reviewer’s comment. We corrected the sentence as follows:

„Including preventive agents had no beneficial impact on the rats' weight increase (in groups of DOX+DEX, DOX+CVD, DOX+DEX+CVD compared to control). Interestingly, the incidence of this phenomenon was decreased in these groups compared to the group of rats that received only DOX (Table 1).”

3) In discussion, the following sentence was not clear: “Notably, pancreatic damage linked to diabetes, as well as heart failure and recurrent arrhythmias attributable to cardiac muscle impairment, are sequelae of this phenomenon [35]. However, this scenario materializes when an excessive influx of iron occurs from the gastrointestinal tract.”  (Page 10, line 320-323). Please, rewrite.

Response 3: We agree with the Reviewer. Therefore, according to the Reviewer suggestion, the sentence was corrected as follows: „The occurrence of diabetes is linked to pancreatic damage, whereas heart failure and frequent arrhythmias are the result of damage to the heart muscle [35]. However, this occurs when excessive iron is absorbed from the gastrointestinal tract”

4) I am not agreeing with this sentence in the discussion: “The observed disturbances in weight growth within the groups administered with 329 DOX are attributed to metabolic disruptions rather than a reduction in the intake of food or water.” First, the authors observed a significantly decrease in water intake among groups that received doxorubicin and in food intake in DOX group compared to C group. Second, there are several studies showing that Dox decrease food intake. Third, the authors did not evaluate any metabolic parameter. The discussion at this point was too speculative.

Response 4: We agree with the Reviewer’s comment. We deleted the sentence.

5) Although ascites is commonly related to cardiac or liver damage, in experimental context, when doxorubicin was administered intraperitoneally, other important factor could influence the incidence of ascites: the peritonitis induced by the presence of doxorubicin in the peritoneal cavity. It is important to take into in mind. Additionally, in the present study, the authors did not show results for ascites. I believe it could be removed from the discussion.

Response 5: We agree with the Reviewer’s comment, therefore we added the sentence as follows: „The administration of DOX intraperitoneally may impact the occurrence of ascites: the peritonitis resulting from the presence of DOX in the cavity of the peritoneum. It is crucial to take this into account.”

6) Conclusion is too long and only summarizing the main results. Please, rewrite.

Response 6: We agree with the Reviewer’s comment. We have revised the summary to shorten the text and improve its readability.

7) Design study described 10 experimental groups, but the authors presented results only for 5 groups…

Response 7: We understand the reviewer's concerns. There was 5 study groups. Rats were administered with drugs every week for ten weeks. One week after the end of the drug administration, half of the rats per group were euthanized. The remaining rats were euthanized eleven weeks after the end of the drug administration. Therefore, the creation of subgroups has been mistakenly described. The Experimental design was corrected.

8) Please, correct the RT-PCR acronym. For the real time PCR, the acronym should be qPCR (page 14).

Response 8: We agree with the Reviewer. The acronym was corrected.

9) Did all the variables have a normal distribution? How were tested the normality? And for histological parameters, how was performed the statistical analysis?

Response 9: All the variables had a normal distribution, which was verified using the Shapiro-Wilk test.  Since most of histological abnormalities were seen only in drug-exposed groups, the incidence was evaluated by chi2 test but due to low number of evaluated animals the result was explained descriptively. In addition, the number of animals in the group in which a change in the indicated severity was observed has been added to the table.

10) The expected results were only observed in histological parameters, which depends on the examinator. It is extremely important detailed the analysis. Is the proficient pathologist blind regarding the groups? How many slides were evaluated? How many histological fields were evaluated?

Response 10: The histological parameters were evaluated by blinded examinator. From one liver sections of two different lobes for each staining were evaluated in each animal (20 slides of HE per group; 20 for PAS per group). Since no morfometric analysis was performer the whole section was evaluated. This facts were added to methods section.

11) Please, update the references.

Response 11: When recalling historical data, it is difficult to refer to current publications. Nevertheless, we have updated the references, as the Reviewer suggested.

12) The tittle should not contain “systemic toxicity”. There is no parameter evaluated regarding systemic toxicity, in my point of view.

Response 12: We agree with the Reviewer. The ‘systemic toxicity’ was removed from the title.

13) The last consideration is related to controverse results related to oxidative stress. We know how difficult is published negative results, mainly among experimental studies. For this reason, we try hardly explaining our results based in other positive results present in scientific literature. The mechanisms involved in doxorubicin toxicity are multiples and could act differently between models (acute, subacute, chronic etc.) and moments of side effects onset. Despite it could be difficult to explain the oxidative stress, I think we only need to take on the results.

Response 13: We agree with the Reviewer. Please indicate the controversial fragment, and we will remove it.

We thank the Reviewers for their thoroughness and the time to review this manuscript. We hope that thanks to their comments and suggestions, and our corrections, the manuscript will be able to be published soon.

Reviewer 2 Report

Comments and Suggestions for Authors

There were some concerns regarding this manuscript.

1.  Cardiac function was not measured.  This is essential given the link the authors are making with Dox induce cardiac dysfunction.

2.  The description for Table 1 and 2 was not very clear about the effect of treatments of weight gain.   Was it all secondary to decreased food intake?

3. No histology analyses were shown for the heart.

4. Validation of gene expression should be performed at protein level.

5.  Differences were not obvious in the PAS staining sections.  Should be quantified.

6.  Immune infiltration should be verified by immunostaining.

Comments on the Quality of English Language

No major concerns.

I would suggest changing decrement to decrease in line 94.   Line 128 was unclear what was meant and could be deleted.

Author Response

Point-by-point response to the reviewer comments

We have read and carefully considered all the comments. We would like to thank the Reviewer for the constructive suggestions and we hope that our responses meet their expectations.

In the corrected version, we have highlighted (with ’tracked changes’) new sentences, rewritten parts and some other changes. Some of these changes were made because of the comments of another reviewer.

Below we provide our point-by-point response to the Reviewer comments and concerns.

Reviewer 2:

  1. Cardiac function was not measured.  This is essential given the link the authors are making with Dox induce cardiac dysfunction.

Response 1: We understand the Reviewer’s point of view, however this work did not focus on the myocardium. Our previous work using the same rats focused on the h=-eart muscle https://pubmed.ncbi.nlm.nih.gov/37373350/

  1. The description for Table 1 and 2 was not very clear about the effect of treatments of weight gain.   Was it all secondary to decreased food intake?

Response 2:The average feed consumption was significantly lower only in DOX group. Could the reviewer be more specific about which part is not clear? Is it the title of the table, the results or the interpretation in the discussion?

  1. No histology analyses were shown for the heart.

Response 3: Data on cardiac dysfunction have been published previously https://pubmed.ncbi.nlm.nih.gov/37373350/

  1. Validation of gene expression should be performed at protein level.

Response 4: The expression of Sod2 and Cat is upregulated in response to oxidative stress mainly at the transcriptional level. When cells experience an increase in ROS levels, it triggers signaling pathways that activate transcription factors like Nrf2, leading to increased expression of these enzymes. In our research, we were primarily interested in the activation of stress-related signaling pathways rather than proteins as the end product. Changes in mRNA levels can indicate alterations in the activity of transcription factors and signaling pathways that control the expression of Sod2 and Cat. Post-transcriptional and post-translational regulatory processes can influence protein abundance independently of mRNA levels. However, it would undoubtedly be necessary to examine protein levels if we were studying changes in gene expression to answer the question of how much a change in RNA levels translated into changes in protein levels.

  1. Differences were not obvious in the PAS staining sections.  Should be quantified.

Response 5:In case of glycogen accumulation a slightly irregular staining distribution was considered as minor intensity change, a small and large focal changes were selected as moderate and major intensity, respectively.

  1. Immune infiltration should be verified by immunostaining. 

Response 6: We agree with the Reviewer that immunostaining helps evaluate the inflammatory infiltration. However, such infiltration has typical histological signs even in HE slides, especially in parenchymal organs such as the liver and near necrotic or perivascular areas.

We thank the Reviewers for their thoroughness and the time to review this manuscript. We hope that thanks to their comments and suggestions, and our corrections, the manuscript will be able to be published soon.

Round 2

Reviewer 1 Report

Comments and Suggestions for Authors

I would like the authors for all the answers. However, I have had some considerations yet.

1) Regarding evaluation of body weight, I believe the authors should choose a statistical test for repeated measures.

2) The authors answered that histological findings were compared by chi-square. Please, include this information in statistical analysis section.

3) I insist that the Conclusion should be rewrite. Please, remove the information about ascites from conclusion. It is acceptable include some discussion about it, but the manuscript results did not support the conclusion.

Author Response

A point-by-point response to the reviewer's comments

We have read and carefully considered all the comments. We would like to thank the Reviewer for the constructive suggestions, and we hope that our responses meet their expectations.

In the corrected version, we have highlighted (with ’tracked changes’) new sentences, rewritten parts, and some other changes. Some of these changes were made because of the comments of another reviewer.

Below, we provide our point-by-point response to the Reviewer's comments and concerns.

Reviewer 1:

  • Regarding evaluation of body weight, I believe the authors should choose a statistical test for repeated measures.

Response 1: We agree with the Reviewer’s comment. We re-analyzed the data using one-way ANOVA for repeated measurements with a Tukey HSD post hoc test. The result of the analysis was not significant for the main conclusions of the study. Appropriate changes were made to the table and text, as well as to the Method section.

  • The authors answered that histological findings were compared by chi-square. Please, include this information in statistical analysis section.

Response 2: The information was included

  • I insist that the Conclusion should be rewrite. Please, remove the information about ascites from conclusion. It is acceptable include some discussion about it, but the manuscript results did not support the conclusion.

Response 3: The conclusions were corrected.

We thank the Reviewer for their thoroughness and the time to review this manuscript. We hope that thanks to their comments and suggestions and our corrections, the manuscript will be able to be published soon.

Reviewer 2 Report

Comments and Suggestions for Authors

No further concerns

Author Response

We thank the Reviewer for their thoroughness and the time to review this manuscript. We hope that thanks to their comments and suggestions, and our corrections, the manuscript will be able to be published soon.